# Establishment, Genetic Diversity, and Habitat Suitability of *Aedes albopictus* Populations from Ecuador

**DOI:** 10.3390/insects13030305

**Published:** 2022-03-19

**Authors:** Andrés Carrazco-Montalvo, Patricio Ponce, Stephany D. Villota, Emmanuelle Quentin, Sofía Muñoz-Tobar, Josefina Coloma, Varsovia Cevallos

**Affiliations:** 1Instituto Nacional de Investigación en Salud Pública, Gestión de Investigación, Desarrollo e Innovación, Quito 170136, Ecuador; andres.carrazco@hotmail.com (A.C.-M.); wpponcey@yahoo.com (P.P.); stephany.villota.v@gmail.com (S.D.V.); munoztobarsofia@gmail.com (S.M.-T.); 2Centro de Investigación en Salud Pública y Epidemiología Clínica Centro, Universidad UTE, Área Geomática-Matemática, Quito 170129, Ecuador; emmanuelle.quentin@gmail.com; 3Sección Invertebrados, Instituto Nacional de Biodiversidad, Quito 170135, Ecuador; 4Division of Infectious Diseases and Vaccinology, School of Public Health, University of California Berkeley, Berkeley, CA 94720-7360, USA; colomaj@berkeley.edu

**Keywords:** *Aedes albopictus*, haplotype, COI, Ecuador

## Abstract

**Simple Summary:**

The tiger mosquito, *Aedes albopictus*, is a vector of multiple viral diseases. Therefore, it is crucial to understand its distribution and the genetic diversity of Ecuadorian populations. This study used a genetic marker to understand aspects of the dynamics of the Ecuadorian populations of *Aedes albopictus* from five sites along the coastal, Amazon basin, and Andean lowland regions. Our results evidence two haplotypes within the Ecuadorian populations. Haplotype 1 was found in the coastal regions and Amazon basin, while haplotype 2 was found in the northeastern lowlands. Worldwide, haplotype 1 has been found in 21 countries in temperate and tropical habitats, and haplotype 2 has been found in five countries in tropical habitats. Hence, a difference in adaptation traits could be speculated between both haplotypes. Our study presents a prediction model that shows the suitable habitats for *Aedes albopictus* in Ecuador. Our results showed that the vector could spread through all country regions, including the Galapagos Islands. Thus, understanding the different aspects of the local populations can help establish better vector surveillance and control strategies.

**Abstract:**

*Aedes albopictus*, also known as the tiger mosquito, is widespread worldwide across tropical, subtropical, and temperate regions. This insect is associated with the transmission of several vector-borne diseases, and, as such, monitoring its distribution is highly important for public health. In Ecuador, *Ae. albopictus* was first reported in 2017 in Guayaquil. Since then, the vector has been identified in the Northeastern lowlands and the Amazon basin. This study aims to determine the genetic diversity of Ecuadorian populations of *Ae. albopictus* through the analysis of the mitochondrial gene COI and to describe the potential distribution areas of this species within the country. The genetic diversity was determined by combining phylogenetic and population genetics analyses of five localities in Ecuador. Results showed two haplotypes in the Ecuadorian populations of *Ae. albopictus*. Haplotype 1 (H1) was found in the coastal and Amazon individuals, while haplotype 2 (H2) was only found in the three northeastern lowlands sites. In a worldwide context, H1 is the most widespread in 21 countries with temperate and tropical habitats. In contrast, H2 distribution is limited to five countries in tropical regions, suggesting fewer adaptation traits. Our prediction model showed a suitable habitat for *Ae. albopictus* in all regions (coastal, Amazon basin, and Andean lowland regions and the Galápagos Islands) of Ecuador. Hence, understanding different aspects of the vector can help us implement better control strategies for surveillance and vectorial control in Ecuador.

## 1. Introduction

Ecuador is located on the northwest coast of South America, and its territory includes the continental mainland and the Galápagos Islands in the Pacific Ocean. The mainland of Ecuador is divided geographically into coastal lowlands in the west of the Andes Mountains, the Andes or highlands, with valleys and mountains with snowed peaks of up to 6300 m, and lowlands on the east of the Andes that are part of the Amazon basin.

*Aedes albopictus (*Skuse, 1894*)*, commonly known as the tiger mosquito, is native to Southeast Asia and is considered one of the most dangerous exotic species [1]. The first record of this species was reported in South Korea in 1940. Nowadays, *Ae. albopictus* is a widespread mosquito species across tropical, subtropical, and temperate regions of the world [2]. Along with *Ae. aegypti*, *Ae. albopictus* is classified as highly important in transmitting vector-borne diseases [1,3] and is associated with the transmission of several arboviruses, such as the Dengue Virus (DENV), Japanese Encephalitis (EJV), Eastern Equine Encephalitis (EEV), Venezuelan Equine Encephalitis (VEEV), St. Louis Encephalitis (SLE), West Nile Virus (WNV), and Yellow Fever (YFV) [1,4,5]. It is also incriminated as a vector for *Dirofilaria immitis*, a parasite of veterinary importance [6]. The wide distribution of this species is often associated with its tolerance to low temperatures and egg resistance to desiccation [2].

In Ecuador, *Ae. albopictus* was first reported in 2017 in Guayaquil [7]. Since then, the species has been recorded in new locations in the Northeastern lowlands and the Amazon basin. However, there is no information regarding the genetic identity of these mosquito populations. The establishment of this invasive mosquito species and climate change can increase the risk of disease transmission in tropical and subtropical areas in Ecuador.

Generally, molecular markers such as allozymes, nuclear, or mitochondrial genes are used to study diversity or inter and intraspecies genetic differences [8]. A recently published study compared the use of cytochrome c oxidase subunit 1 (COI) and the 18s gene, exploring the genetic variability of each, showing a higher success rate for COI and a more significant potential for species identification compared to the ribosomal marker [9]. Among the advantages of using the COI gene are its lower variability in the total length and the metrics based on the variability by codon position or the detection of stop codons [10].

Mitochondrial genes, such as COI, are widely used to understand vector insect populations and their molecular, genetic evolution, dispersal patterns, phylogeny, and population dynamics [11,12]. These conserved genes present high levels of polymorphism and divergence among species, making them good molecular markers [13,14] that can help describe pest populations to carry out adequate entomological monitoring [15] and implement better control practices.

This study aims to determine the genetic diversity of *Ae. albopictus* in the Ecuadorian territory through the analysis of the mitochondrial gene cytochrome c oxidase subunit 1 (COI), using a combination of phylogenetic and population genetic analyses. Furthermore, we aim to present the potential distribution areas for this species using maximum entropy models to predict suitable areas for its dispersion.

## 2. Materials and Methods

### 2.1. Sample Collection

Individuals of *Ae. albopictus* were collected between 2017 and 2021 in five localities of Ecuador (Table 1; Appendix A, collecting permit MAE-DNB-CM-2018-0109). Adult individuals were collected using the BG-Sentinel traps without lures in Guayaquil city. Immature individuals were collected from artificial hatcheries (metal pots and plastic gallons) and ovitraps from Puerto Francisco de Orellana, Lita, Guadual, and Cachaco. Adult individuals were preserved dry in the “National Collection of Arthropod Vectors” under the operating patent 018-2019-FAU-DPAP-MA. Immature individuals were transported to the laboratory and emerged as adults. Adult individuals were identified following Rueda’s (2004) pictorial keys to identify mosquitoes.

### 2.2. DNA Extraction, Amplification, and Sequencing of the COI Gene

A total of 51 specimens of *Ae. albopictus* were used for DNA extractions. The small number of individuals used for the genetic analyses is due to the limited sampling of an ongoing dispersion of the mosquito.

Depending on the number of specimens per site, samples were obtained from either legs or the entire body from fresh material or dry specimens. Genomic DNA was extracted using the DNeasy Blood & Tissue Kit^®^ (Qiagen, Hilden, Germany), following the manufacturer’s instructions.

Amplification of mitochondrial cytochrome *c* oxidase subunit I (COI) was done with the primers LCO1490 (5′-GGTCAACAAATCATAAAGATATTGG-3′) and HC02198 (5′-TAAACTTCAGGGTGACCAAAAAATCA-3′) [16]. Each reaction contained 1× DreamTaq Green PCR Master Mix (Thermo Scientific™ K1081, Waltham, MA, USA) and 400 nM of each primer. Reactions were done in 25 µL volume with 5 µL of DNA. The cycling amplification profile was: three minutes at 95 °C, 35 cycles of one minute at 95 °C, one minute at 45 °C, and one minute at 72 °C, followed by a final extension of 72 °C for seven minutes. PCR products were detected by 2% agarose gel electrophoresis in TAE buffer, stained using SYBR^®^ Safe 10000X. PCR products were sequenced using the Sanger sequencing method at Macrogen, Seoul, South Korea.

### 2.3. Phylogenetic Analyses

#### 2.3.1. Haplotypes Network

We determined the number of circulating haplotypes in Ecuadorian populations using PopART [17] and DnaSP with default settings [18]. Likewise, the haplotypes were mapped using the QGIS 3.10.0 software [19].

To determine the origin of the Ecuadorian haplotypes, we downloaded 2261 COI gene sequences of *Ae. Albopictus*, available in the GenBank-NCBI database. Sequences in this portion of the analyses included individuals from America, Europe, Africa, and Asia. We constructed a preliminary haplotype network using the entire data set (PopArt) to establish the sequences close to the Ecuadorian haplotypes (Appendix A). This information was also used to assemble a world distribution map of haplotypes.

#### 2.3.2. Phylogenetic Tree

Phylogenetic analyses were performed to compare the Ecuadorian haplotypes with previously recorded COI haplotypes across the globe. For this analysis, we chose 57 sequences of up to 1525 bps of different haplotypes distributed around the world. These sequences have been used in different prior studies (Appendix A). Sequences of *Culex quinquefascitus* (Say, 1823) and *Ae. aegypti* (Linnaeus, 1762) were used for family-level outgroup representation. Voucher specimens of this study are deposited in the “National Collection of Arthropod Vectors” at the National Institute of Research in Public Health—Dr. Leopoldo Izquieta Perez (INSPI-Quito). Phylogenetic analyses were performed in MrBayes on XSEDE [20] and RAxML-HPC BlackBox [21] through CIPRES Science Gateway v.3.3 (phylo.org) [22]. The analyses were performed using a GTR + G + I model, with two independent runs through 1,000,000 generations; trees were sampled every 100th generation. While RAxML analyses included 1000 bootstrap replicates using default settings. Lastly, consensus trees were generated in PAUP4 [23], using a 50% majority rule.

### 2.4. Genetic Diversity and Population Structure

The genetic diversity and population structure of the five Ecuadorian populations of *Ae. albopictus* were analyzed using default settings in DnaSP [18] and Arlequin 3.5 [24]. To determine the genetic diversity of the populations, we calculated the following indexes: haplotype diversity (h), nucleotide diversity (π), number of polymorphic sites (S), and number of migrants per generation (Nm). We determined whether populations evolve under a non-random process by calculating the neutrality test Tajima’s D (D) and Fu’s FS (F) using DnaSP. In Arlequin, we calculated FST values to determine the level of genetic connectivity among populations. The analysis of the molecular variance (AMOVA) was used to test the effect of geographical barriers in the *Ae. albopictus* populations (Tests 1 and 3) and compare populations with both haplotypes (Test 2). Furthermore, major geographic barriers appear to be shaping the genetic diversity of other insect lineages [25]. To test this effect, we parsed our data set around the presence of the Andes mountain chain and the major rivers on the coast, such as the Esmeraldas and Guayas rivers (test 3) [26]. We calculated a Mantel Test performed in Arlequin, using 1000 randomizations, to correlate genetic (FST) and geographical distances. We estimated the number of populations (k) in Structure 2.3.4 [27] using a no-admixture model and ten iterations per run, with five viable populations in our analyses.

### 2.5. Maximum Entropy Model of the Asian Tiger Mosquito in Ecuador

In order to apply the maximum entropy model Maxent [28], the worldwide locations described for *Ae. Albopictus* presence were downloaded from the Global Biodiversity Information Facility (GBIF, 2021) [29]. The points within the American continent, mainly concentrated in Brazil and southeast of the United States, were used for the analysis (22,142 in total) (Figure 1).

The explicative variables consisted of the following rasters obtained by satellite images processing: elevation layer and 2005–2019 monthly climatic variables (EpiClim database), including average humidity, total precipitation, average daily temperature, average nightly temperature, and average vegetation index.

The Maxent version 3.4.4 was used with the default option of the cloglog unit. This unit gives an estimate between 0 and 1 of the probability of presence.

## 3. Results

### 3.1. Phylogenetic Analyses

#### 3.1.1. Haplotypes Network

The COI data set for *Ae. albopictus* spanned 684 bps, with two parsimony informative sites. Haplotype designation revealed two haplotypes in the Ecuadorian populations of *Ae. albopictus* (Figure 2, GenBank accessions available in Appendix A). Haplotype 1 (H1) included recorded specimens of the coastal city of Guayaquil and from Puerto Francisco de Orellana, located in the Amazonian region. In contrast, haplotype 2 (H2) was found in three lowland sites surveyed in the province of Imbabura (Lita, Cachaco, and Guadual). Haplotype network analyses showed no shared haplotypes among populations of the Imbabura province and individuals from Guayaquil and Puerto Francisco de Orellana (Figure 2, Appendix A). The two haplotypes reported in Ecuadorian populations vary in two nucleotides (Figure 2).

In a global context, when we compared the haplotypes present in Ecuador with sequences available in GenBank, H1 was found to be the most widespread haplotype; it is present in the Americas (Ecuador, Canada, United States, Panama [30], Brazil), Europe (Spain, Portugal, Italy, Russia, Bulgaria, Austria, Czech Republic, Greece, Montenegro, Turkey, Albania [31]), Africa (Morocco, Mauritius [32]), and Asia (China, South Korea [33], Japan, Pakistan), while H2 is present in the tropical equatorial countries of South America (Ecuador, Colombia), Africa (Cameroon, Congo), and Asia (Singapore [34], Malaysia) (Figure 3).

#### 3.1.2. Phylogenetic Tree

The phylogenetic tree displays one representative per country in each cluster. Sequences from China, Pakistan, and Malaysia are closer to the defined outgroups. Phylogenetic analyses performed with the mitochondrial marker using Bayesian and maximum likelihood support the *Ae. albopictus* clade through bootstrap and posterior probability (Figure 4). This clade presented 10 well-defined clusters, with support for posterior probabilities and bootstrap values ranging between 0.5 and 1.0 (Figure 4). The Ecuadorian H1 is located at the base of the clade, with individuals from 25 countries, while Cluster 9 grouped individuals from five countries (including Colombia) and the Ecuadorian H2.

### 3.2. Genetic Diversity and Population Structure

*Aedes albopictus* populations in Ecuador showed a low nucleotide diversity (π = 0.0142). These results were supported by the haplotype diversity of the sampled sites. Only one population was detected in the structure analyses (K = 1; Appendix A), and Ecuadorian populations do not show selection through neutrality test values (D = 0, F = 0). However, overall FST values (FST = 0.20–0.60) showed moderate genetic differentiation among these five populations. When we compared FST values on a population basis, the highest differentiation was observed when Puerto Francisco de Orellana individuals were compared to Lita, Cachaco, and Guadual samples (FST = 0.54–0.60, Table 2). This differentiation is consistent with regional representation. In contrast, moderate differentiation was also reported between samples from Guayaquil and Lita, Cachaco, and Guadual (FST = 0.29–0.33, Table 2). This moderate differentiation is consistent with the geographical distance between sites, although the Mantel test showed no significant correlation between genetic and geographical distances (*p* = 0.15, R^2^ = 0.25). No migrants were detected among sites, and most of the variation in the AMOVA was found within groups when testing for regional representation and haplotype distribution (Table 3).

### 3.3. Maximum Entropy Model of the Asian Tiger Mosquito in Ecuador

The model generated with present bioclimatic variables predicts the presence of the Asian tiger mosquito in the coastal areas, Galápagos Islands, lowland areas of the Andes, and the Ecuadorian Amazon basin (Figure 5a). The model’s performance is evaluated by the AUC (area under the curve) from the ROC (receiver operating characteristics) curve, giving an ROC value of 0.756 (Figure 5b), being an optimal model as it is close to 1.0 [35].

## 4. Discussion

The first reported record of *Aedes albopictus* in Ecuador was found in Guayaquil (Pacific Coast) in 2017 [7]. From this event, different entomological samplings have been carried out to monitor the dispersion of the insect in the country. We report an expanded distribution of *Ae. albopictus* in the Amazon basin (Francisco de Orellana) and localities in the northwestern Andean lowlands (Cachaco, Lita, and Guadual) at the Imbabura province. The dispersion of this species in Ecuador is ongoing since we have not recorded the mosquito in several lowland locations that are constantly monitored (Cevallos, Unpublished data).

The haplotype network showed two different genetic groups (Figure 2). H1 includes individuals from the Pacific coast (Guayaquil) and the Amazon basin (Puerto Francisco de Orellana), separated by 605 km. As suggested in other invaded areas [36], the latter locality may have been colonized through human activities. In contrast, H2 was distributed in three locations in the northwestern Andean lowlands. *Ae. albopictus* was initially found in Cachaco, and it expanded to Lita and Guadual. These three sites are located on a road that connects Ecuador and Colombia. This geographical connection may help understand the similarities found in this work between Ecuadorian *Ae. albopictus* H2 and specimens from Colombia. Therefore, our results showed the possibility of more than one introduction and colonization event in the country. The global haplotype network (Figure 3) indicates that Ecuadorian H1 is distributed in 21 countries in America, Europe, Africa, and Asia. H1 presents a worldwide distribution, and, as such, these individuals show adaptation traits to occupy temperate [37] and tropical habitats. In contrast, H2 is distributed in the tropical regions of five countries in America (Colombia: 7°0′0″ N, 75°30′0″ O), Africa (Cameroon: 7°22′10.999″ N, 12°21′16.999″ E; Congo: 4°19′39.288″ S, 15°18′48.852″ E), and Asia (Singapore: 1°17′24.972″ N, 103°51′7.052″ E; Malaysia: 3°8′27.071″ N, 101°41′35.545″ E). This restricted distribution could indicate that H2 has not adapted to climatic conditions far from the equator. These results agree with a previous report in which, after analyzing the spatial, temporal, and genetic invasion of *Ae. albopictus*, the insect’s probability of expansion and adaptation to cold and winter climates was reported as low [38]. However, over time, the continued propagule pressure and establishment of populations will drive the establishment and expansion of this mosquito outside its usual range.

Several studies have reported more haplotypes for *Ae. albopictus* populations using COI as a molecular marker. Our results showed similarities within our Ecuadorian haplotypes and worldwide sequences (Figure 4). For instance, Ecuadorian H1 is similar to the newly reported haplotype in Panamá (H72) [30], Albania (A1a2a1) [31], South Korea (H28) [33], and Singapore (H32) [34]. Likewise, our H2 is similar to Singapore (H27) [34]. Due to the difference in lengths of our sequence compared to these reports, we can only suggest these haplotypes as similar. A more extended COI sequence would help determine the exact relationship within *Ae. albopictus* haplotypes that are reported worldwide. Furthermore, phylogenetic analyzes showed that Ecuadorian H1 (Figure 4) is the most widespread haplotype. Hence, this haplotype may be part of the initial populations that later emerged and spread worldwide. On the other hand, Ecuadorian H2 (Figure 4) has a limited distribution. It is necessary to corroborate these genomic changes with whole-genome studies to analyze the distribution of SNPs in the species. According to the study by Schmidt and collaborators [39], currently, there is no specific understanding of the dispersal processes of *Ae. albopictus* due to the absence of studies using high-resolution genetic markers. Their results show that by analyzing SNPs in large regions of the genome or complete genomes, robust results can be obtained on gene flow and genetic differentiation in invasion and dispersion processes [39].

Ecuadorian populations of *Ae. albopictus* presented reduced genetic variability, which was supported by STRUCTURE and AMOVA. These analyses showed the absence of population structure (K = 1) and that most genetic variation was within populations (Table 3). Even though these populations present low genetic diversity, some genetic differentiation was found when comparing individual FST values (Table 2). High levels of genetic differentiation were recorded when the three rural sites in the Andean lowlands were compared to the site in the Amazon region (FST = 0.54–0.60, Table 2). We also found moderate levels of genetic differentiation between Guayaquil and the four other sites analyzed (FST = 0.2–0.30, Table 2), which are geographically distant. However, the Mantel test found no correlation between genetic and geographical distances. Nonetheless, considering the small number of populations analyzed and the time since the introduction of the vector to Ecuador, these results must be considered preliminary. Other studies comparing insect populations have found little evidence of genetic structure even among distant populations, suggesting that dispersal along human transportation networks is expected. The success of *Ae. albopictus* as an invasive species is based on its dispersal capacity in human transport networks, specifically on roads and navigation [40,41]. Continued surveillance and sampling in the main ports of entry and around the known mosquito populations will provide us with a better understanding of the vector dynamics in the country.

Studies predict that climate change and human mobility will promote the spread of the tiger mosquito, increasing the risk of pathogen transmission affecting human health [6,42]. Furthermore, the establishment of *Ae. albopictus* depends on weather abnormalities, urbanization, and socio-economic factors [43]. Our study used niche modeling to predict suitable areas for *Ae. albopictus* in the Ecuadorian territory using present bioclimatic variables [44]. Suitable habitat for the vector includes all four regions of Ecuador, particularly in areas of the Pacific coast, Galapagos Islands, and the Amazon region, with a portion of suitable habitat in the Andean lowlands (Figure 5a). Our model showed a larger area of the predicted distribution of the Ecuadorian territory than previous models [45]. This difference is probably due to local sites not being included in the previous models. The most relevant variables for the model’s training appear to be humidity, followed by the vegetation index, precipitation, and nightly temperature. The understanding of the vector genetic diversity and other factors that promote its establishment and distribution can help us implement better control strategies for the surveillance and control of *Ae. albopictus* in Ecuador.

## Figures and Tables

**Figure 1 insects-13-00305-f001:**
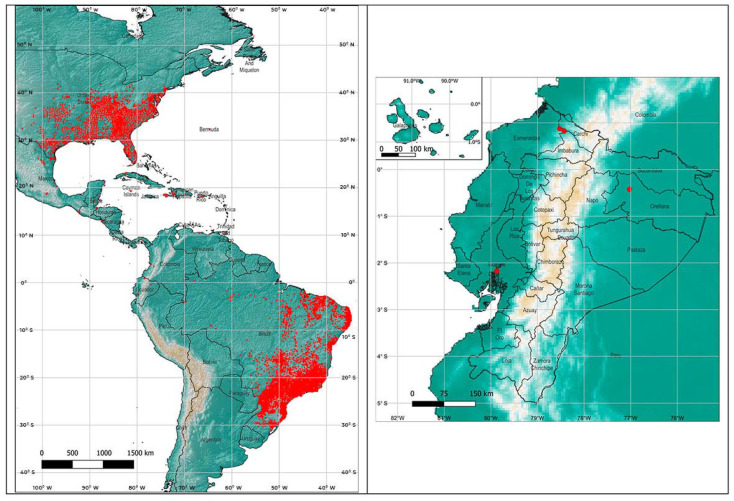
Presence points of *Aedes albopictus* in the Americas, obtained from the GBIF (2021) and presence points in Ecuador.

**Figure 2 insects-13-00305-f002:**
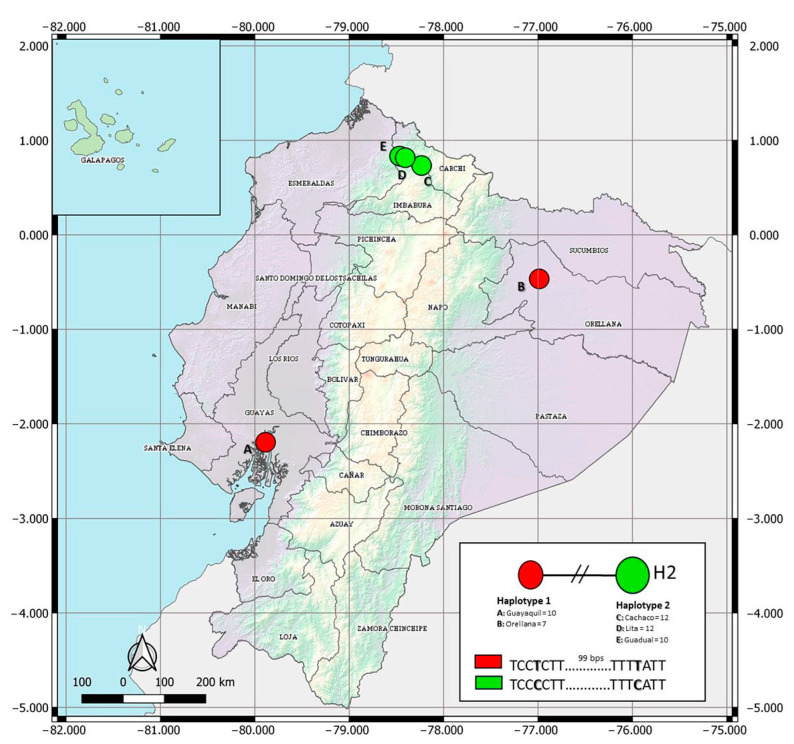
Haplotype network of the COI gene for Ecuadorian samples of *Aedes albopictus*. H1 represents 17 sequences, 10 from Guayaquil and 7 from Orellana. H2 represents 34 sequences, 12 from Cachaco, 12 from Lita, and 10 from Guadual.

**Figure 3 insects-13-00305-f003:**
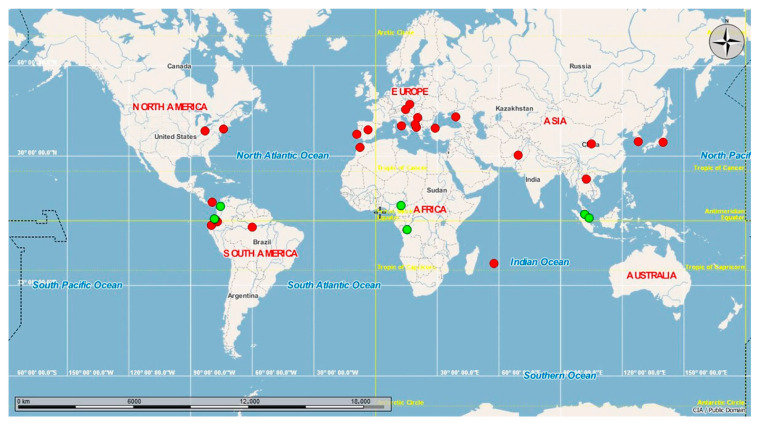
Global haplotype network. Worldwide distribution of *Aedes albopictus* haplotypes reported in Ecuador.

**Figure 4 insects-13-00305-f004:**
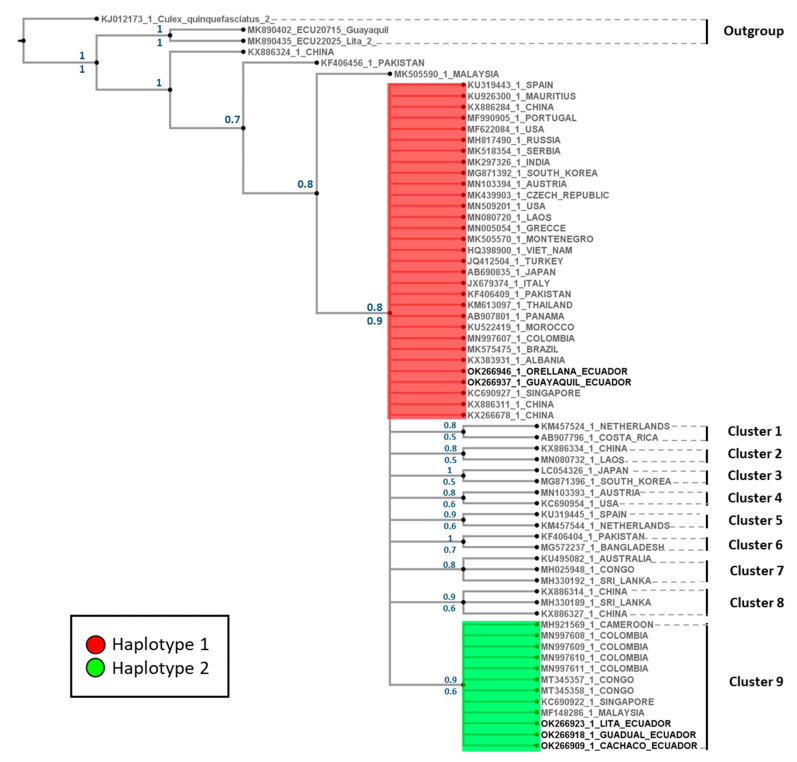
Bayesian 50% rule consensus tree for *Aedes albopictus* using the COI gene. Posterior probabilities are shown above the branches, and bootstrap support values for the ML tree are shown below the branches.

**Figure 5 insects-13-00305-f005:**
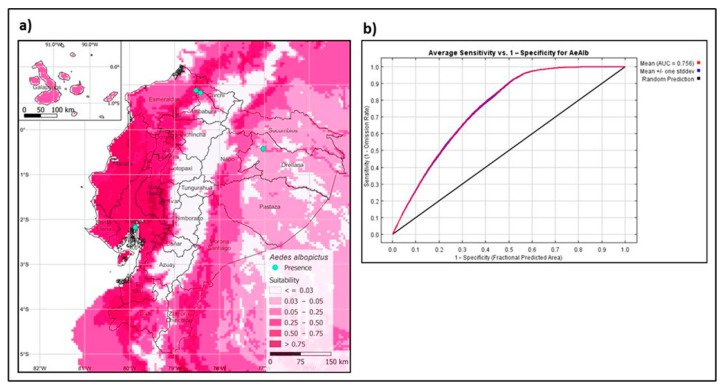
Maximum entropy models for the present distribution of *Aedes albopictus*. (**a**) Spatial distribution of *Ae. albopictus* in Ecuador. (**b**) Receiver operating characteristics with AUC value.

**Table 1 insects-13-00305-t001:** Collecting sites for *Aedes albopictus* individuals in tropical and subtropical regions of Ecuador.

Region	Province	Locality	Latitude	Longitude	Collection Year	Sample Size
Andean Lowland	Imbabura	Lita	0.8337	−78.4017	2018–2020	12
	Imbabura	Guadual	0.8918	−78.5022	2021	10
	Imbabura	Cachaco	0.8337	−78.4023	2018–2019, 2021	12
Pacific coast	Guayas	Guayaquil	−2.1633	−79.8938	2017	10
Amazon basin	Orellana	Francisco de Orellana	−0.4412	−77.0048	2018–2020	7

**Table 2 insects-13-00305-t002:** FST values for individuals from five populations of *Aedes albopictus* from Ecuador. *p*-value ^a^
*p* < 0.001.

	Guayaquil	Puerto Francisco de Orellana	Lita	Cachaco	Guadual
Guayaquil	0				
Puerto Francisco de Orellana	0.20	0			
Lita	0.29 ^a^	0.54 ^a^	0		
Cachaco	0.33 ^a^	0.60 ^a^	−0.01	0	
Guadual	0.30 ^a^	0.56 ^a^	−0.02	−0.11	0

**Table 3 insects-13-00305-t003:** Analysis of the molecular variance for five Ecuadorian populations of the Asian tiger mosquito.

Number of Groups	Partitions	Test	Among Groups	Among Populations	Within Groups
2	(1,2,3,4) (5)	Coast vs. Amazon basin	38.35	11.26	50.39
2	(1,2,3) (4,5)	Haplotype 1 vs. haplotype 2	39.87	3.74	56.38
3	(1,2,3) (4) (5)	NW coast, SW Coast and Amazonia	45.46	−2.30	56.85

## Data Availability

Sequences used in this study are deposited in GenBank under the accession numbers OK266899–OK266949.

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
