# Peer review of "Establishment, Genetic Diversity, and Habitat Suitability of Aedes albopictus Populations from Ecuador"

_insects, 2022, doi:10.3390/insects13030305_

Round 1

Reviewer 1 Report

In general this is a nice written paper on the genetic diversity of Ecuadorian populations of Aedes albopictus based on COI sequencing. However, I pointed out some minor issues that should be addressed before the manuscript is ready for publication.

1- The introduction is too much concise. The authors should give more information regarding the suitability of mitochondrial DNA for such molecular evolution and population genetics studies. Moreover, there are plenty of examples in the literature that report vector populations genetics studies using COI sequencing. Please, consider to give some of them.

2- Please explain in more detail how AMOVA was used to test the effect of geographical barriers (mountain ranges, dry valleys, and major rivers) like stated on M and M section

3- The sample size for this kind of study is a little bit bellow the average. Usually, Aedes albopictus specimens are quite easy to collect. This specific point is not a limiting one, (as can easily be seen in the Fst differences between populations, for instance) but the low numbers of specimens should be explained

4-The authors mention an analysis using the STRUCTURE software, but there is no results of this analysis presented in the manuscript as a graph. I understand that there was only 1 genetic background found, but the graph should be presented as a supplementary figure.

5- The discussion is too much descriptive. Please improve it taking into account the results that you present, compared to myriad of examples of such studies in mosquito vectors

Author Response

Response to Reviewer 1 Comments

In general this is a nice written paper on the genetic diversity of Ecuadorian populations of Aedes albopictus based on COI sequencing. However, I pointed out some minor issues that should be addressed before the manuscript is ready for publication.

Point 1: The introduction is too much concise. The authors should give more information regarding the suitability of mitochondrial DNA for such molecular evolution and population genetics studies. Moreover, there are plenty of examples in the literature that report vector populations genetics studies using COI sequencing. Please, consider to give some of them.

Response 1: Additional information regarding the use of COI as a molecular marker was added to the introduction.

Point 2: Please explain in more detail how AMOVA was used to test the effect of geographical barriers (mountain ranges, dry valleys, and major rivers) like stated on M and M section

Response 2: This was addressed in the manuscript. The M and M section describes which geographical barriers are being tested (the Andes and significant rivers on the Coast).

Point 3: The sample size for this kind of study is a little bit bellow the average. Usually, Aedes albopictus specimens are quite easy to collect. This specific point is not a limiting one, (as can easily be seen in the Fst differences between populations, for instance) but the low numbers of specimens should be explained

Response 3: Before this study, Ae. albopictus has only been reported in Guayaquil city. Now we report the insect in other locations: Amazon and Andes. Based on our records of ongoing research held in different locations, Ae. albopictus is not very widespread in Ecuador. Individuals in a similar number of these localities were chosen according to availability in the National Collection of Vector Arthropods at INSPI and economic resources for sequencing.

Point 4: The authors mention an analysis using the STRUCTURE software, but there is no results of this analysis presented in the manuscript as a graph. I understand that there was only 1 genetic background found, but the graph should be presented as a supplementary figure.

Response 4: A triangle plot and a bar plot corresponding to the STRUCTURE analysis have been added to the supplementary figures (Figures S1 and S2, accordingly).

Point 5: The discussion is too much descriptive. Please improve it taking into account the results that you present, compared to myriad of examples of such studies in mosquito vectors

Response 5: Additional information regarding other studies was added to the discussion section.

Reviewer 2 Report

The tiger mosquito, Aedes albopictus, invaded in Ecuador in 2017 and is now found in diverse geographical regions of the country. Authors in the present paper show that two haplotypes of the tiger mosquito exist in Ecuador, H1 in the coastal and Amazon regions, and H2 in three northeastern lowland sites. The study must be considered as preliminary because only 51 specimens from 5 locations were sampled and analyzed.  Moreover, the paper is mainly of local interest. Nevertheless, I suggest to publish the manuscript because the tiger mosquito represents a dangerous invasive species for humans in its function as vector for many serious deseases.

The experiments are carefully done and the manuscript is well written. It needs only minor corrections:

(1) line 25 should be separated into two sentences

(2) line 172: network in lowercase letter

(3) line 200: outgroups in lowercase letter

(4) line 224: the sentence beginning with "although" is not correct

(5) Table 2: explain what "c" means

(6) The references must be written in a uniform style and according to MDPI Authors' Instructions (e.g., title of papers in lowercase letters; species names in italics)

Author Response

Response to Reviewer 2 Comments

The tiger mosquito, Aedes albopictus, invaded in Ecuador in 2017 and is now found in diverse geographical regions of the country. Authors in the present paper show that two haplotypes of the tiger mosquito exist in Ecuador, H1 in the coastal and Amazon regions, and H2 in three northeastern lowland sites. The study must be considered as preliminary because only 51 specimens from 5 locations were sampled and analyzed.  Moreover, the paper is mainly of local interest. Nevertheless, I suggest to publish the manuscript because the tiger mosquito represents a dangerous invasive species for humans in its function as vector for many serious deseases.

The experiments are carefully done and the manuscript is well written. It needs only minor corrections:

Point 1: line 25 should be separated into two sentences

Response 1: Thank you for the comment. We agree with the change.

Point 2: line 172: network in lowercase letter

Response 2: Thank you for the comment. We agree with the change.

Point 3: line 200: outgroups in lowercase letter

Response 3: Thank you for the comment. We agree with the change.

Point 4: line 224: the sentence beginning with "although" is not correct

Response 4: We have revised and made the necessary changes.

Point 5: Table 2: explain what "c" means

Response 5: The differentiation values (FST) between samples from Guayaquil compared to samples from Lita, Cachaco and Guadual showed a significant difference (p < 0.001). Changes were made accordingly through the table.

Point 6: The references must be written in a uniform style and according to MDPI Authors' Instructions (e.g., title of papers in lowercase letters; species names in italics)

Response 6: Thank you for the comment. Changes were made accordingly through the references.

Reviewer 3 Report

The paper looks at the invasive mosquito Aedes albopictus which is recently identified in Ecuador. Mosquito specimens were collected in five locations and the mitochondrial COI 5´region was sequenced from 52 specimens. All sequences fall into two haplotypes differing in two basepairs and all locations only report one haplotype. From this sequence information the authors use several statistical models to analyse differences between the locations. However, there is not enough genetic information in the data to warrant that type of analysis. Finally, the paper also presents a model predicting suitable habitats for Aedes albopictus.

I lack a further discussion about the methods used in the paper. While the COI barcoding region is very useful for species identification it is not suited to study intraspecies differences. While COI sequence might occasionally correlate with phenotypic differences such as the temperature tolerance suggested in the paper, such differences would be better studied with multi locus methods such as microsatellites or ddRAD. How well does mitochondrial linages correspond with traits such as temperature adaptation? No previous research is sited.

Given that you do look at COI, how many other haplotypes are present worldwide?

Other methods to study genetic differences are more suited.

Microsatellites:

Latreille, A.C., Milesi, P., Magalon, H. et al. High genetic diversity but no geographical structure of Aedes albopictus populations in Réunion Island. Parasites Vectors 12, 597 (2019). https://doi.org/10.1186/s13071-019-3840-x

ddRAD

Stéphanie Sherpa, Delphine Rioux, Charlotte Pougnet-Lagarde, Laurence Després,

Genetic diversity and distribution differ between long-established and recently introduced populations in the invasive mosquito Aedes albopictus, Infection, Genetics and Evolution,Volume 58, 2018,

FST measurement pointless when only two nucleotides differ. Each location only have one haplotype. Genetic difference is pointless.

“Ecuadorian populations of Ae. albopictus presented reduced genetic variability, which 280 was supported by STRUCTURE and AMOVA. These analyses showed the absence of population structure (K=1) and that most genetic variation was within populations (Table 3).” Looking at your sequences in genbank this variation can only be due to the sequence ends that are not properly truncated. Otherwise you would label them as different haplotypes. Truncating the ends would give you no variation within the subpopulations at all.

Author Response

Response to Reviewer 3 Comments

The paper looks at the invasive mosquito Aedes albopictus which is recently identified in Ecuador. Mosquito specimens were collected in five locations and the mitochondrial COI 5´region was sequenced from 52 specimens. All sequences fall into two haplotypes differing in two basepairs and all locations only report one haplotype. From this sequence information the authors use several statistical models to analyse differences between the locations. However, there is not enough genetic information in the data to warrant that type of analysis. Finally, the paper also presents a model predicting suitable habitats for Aedes albopictus.

Point 1: I lack a further discussion about the methods used in the paper. While the COI barcoding region is very useful for species identification it is not suited to study intraspecies differences. While COI sequence might occasionally correlate with phenotypic differences such as the temperature tolerance suggested in the paper, such differences would be better studied with multi locus methods such as microsatellites or ddRAD. How well does mitochondrial linages correspond with traits such as temperature adaptation? No previous research is sited.

Response 1: This question is very relevant, and we would like to answer it in the future. The analyzed populations in this study have been only found in tropical weather (amazon basin and pacific coast), regions that do not present extreme weather patterns. Additionally, populations of the vector appear to be recently introduced. To answer this question, we need a more extensive sampling, especially in the transition ecosystems of the east and west Andean mountain chains, given that the Asian tiger mosquito presents cold weather resistance in other world regions.

Point 2: Given that you do look at COI, how many other haplotypes are present worldwide?

Response 2: Detailed information was added to the results and discussion sections, and the supplementary data.

Point 3: Other methods to study genetic differences are more suited.

  • Microsatellites: Latreille, A.C., Milesi, P., Magalon, H. et al. High genetic diversity but no geographical structure of Aedes albopictus populations in Réunion Island. Parasites Vectors 12, 597 (2019). https://doi.org/10.1186/s13071-019-3840-x
  • ddRAD: Stéphanie Sherpa, Delphine Rioux, Charlotte Pougnet-Lagarde, Laurence Després, Genetic diversity and distribution differ between long-established and recently introduced populations in the invasive mosquito Aedes albopictus, Infection, Genetics and Evolution,Volume 58, 2018,

FST measurement pointless when only two nucleotides differ. Each location only have one haplotype. Genetic difference is pointless.

Ecuadorian populations of Ae. albopictus presented reduced genetic variability, which 280 was supported by STRUCTURE and AMOVA. These analyses showed the absence of population structure (K=1) and that most genetic variation was within populations (Table 3).” Looking at your sequences in genbank this variation can only be due to the sequence ends that are not properly truncated. Otherwise you would label them as different haplotypes. Truncating the ends would give you no variation within the subpopulations at all.

Response 3: We agree with the reviewer that other molecular markers and techniques are more sensitive to variation and could have been used. Due to a lack of funding, this is not possible at the moment. The methods used in this study to analyze the genetic diversity of Ae. albopictus  (FST, AMOVA, neutral indexes, Mantel test) are standard methods for assessing insect populations. We, the authors, would like to keep the manuscript proposed methods as they are. We hope to incorporate some of these proposed methods in the future and expand the sampling areas of interest.

From our NCBI sequences, it can be seen that the extremes do not affect the variations. The polymorphisms are located at base numbers 357 and 456. At the time of analysis, the gaps at the ends of the shorter sequences are replaced with Ns, therefore the result will not be affected. Regarding the length of the sequences used in this study, there is a higher probability of finding variability among individuals and populations with longer sequences. We, the authors, would like to keep our sequence's length as is.

Round 2

Reviewer 3 Report

It has previously been reported that Aedes albopictus is introduced in Ecuador. The current manuscript extends that to show that it is present in at least five locations in several parts of the country. The COI sequences show that the 52 sequenced specimens are of two haplotypes. Both these haplotypes are present also in other countries which suggests that there have been at least two introductions of Aedes albopictus to the country.

Since the species is a new introduction into the country there is little to suggest that any adaptations to different climates or local environmental factors has occurred after the introduction. Rather the sequences presented are identical to variant present in other locations.

An argument for the H2 haplotype to be more adapted to tropical climates is made with a map of other findings. However, A quick blast made me realize that this haplotype has also been reported also from Los Angeles, USA, (KC690929 and KC690922) Suggesting that it is not only found in tropical regions. The way you present the haplotype as focused in the tropics without including these sequences is misleading and should be changed.

I find that the analysis of the sequence data made in the manuscript is unhelpful to the conclusions that can be drawn from the data. I really do not see the point using statistics to analyse two haplotypes that are so conserved.

My suggestion is that the paper is rewritten to focus on the fact that several introductions of Aedes albopictus has been made to Ecuador and that the species is spreading in several parts of the country. The climate model presented can be used to indicate that the species has the potential to spread further.

Author Response

Response to Reviewer 3 Comments

(Round 2)

It has previously been reported that Aedes albopictus is introduced in Ecuador. The current manuscript extends that to show that it is present in at least five locations in several parts of the country. The COI sequences show that the 52 sequenced specimens are of two haplotypes. Both these haplotypes are present also in other countries which suggests that there have been at least two introductions of Aedes albopictus to the country.

Point 1: Since the species is a new introduction into the country there is little to suggest that any adaptations to different climates or local environmental factors has occurred after the introduction. Rather the sequences presented are identical to variant present in other locations.

Response 1: Precisely, Figure 3 was elaborated to indicate the worldwide distribution of the haplotypes that we reported in Ecuador (H1 and H2). Our article discusses multiple factors  that contributed to the introduction and establishment of the insect vector, emphasizing the establishment of a population under various parameters and favorable environmental factors where the haplotypes have been reported.

Although some studies analyze a larger fragment of the COI gene, several published studies are using only ~600 bps. We provide some examples:

1) K. Futami, A. Valderrama, M. Baldi, N. Minakawa, R. Marín Rodríguez, L. F. Chaves, New and Common Haplotypes Shape Genetic Diversity in Asian Tiger Mosquito Populations from Costa Rica and Panamá, Journal of Economic Entomology, Volume 108, Issue 2, April 2015, Pages 761–768, https://doi.org/10.1093/jee/tou028

2) Maynard, A. J., Ambrose, L., Cooper, R. D., Chow, W. K., Davis, J. B., Muzari, M. O., ... & Beebe, N. W. (2017). Tiger on the prowl: Invasion history and spatio-temporal genetic structure of the Asian tiger mosquito Aedes albopictus (Skuse 1894) in the Indo-Pacific. PLoS neglected tropical diseases, 11(4), e0005546, https://doi.org/10.1371/journal.pntd.0005546,

3) Naim, D. M., Kamal, N. Z. M., & Mahboob, S. (2020). Population structure and genetic diversity of Aedes aegypti and Aedes albopictus in Penang as revealed by mitochondrial DNA cytochrome oxidase I. Saudi journal of biological sciences, 27(3), 953-967. https://doi.org/10.1016/j.sjbs.2020.01.021

In addition to the distribution and model, our significant finding is that there are two different polymorphisms between the haplotypes. These are transmitted and maintained in the generations of a population. We tried to choose sizes limited to ~600 bps, but we did not want to leave out the contribution of some colleagues who sequenced a larger region, focusing only on our fragment. Due to this difference in lengths, we can not affirm that our Ecuadorian haplotypes are identical to those reported in the literature.

Point 2: An argument for the H2 haplotype to be more adapted to tropical climates is made with a map of other findings. However, A quick blast made me realize that this haplotype has also been reported also from Los Angeles, USA, (KC690929 and KC690922) Suggesting that it is not only found in tropical regions. The way you present the haplotype as focused in the tropics without including these sequences is misleading and should be changed.

Response 2:  As the Reviewer points out, the sequences KC690929 (H34) and KC690922 (H27) align with the Ecuadorian H2. However, both sequences are from samples collected in Singapore (Zhong et al. 2013) and not from California, as the Reviewer mentioned. Hence, Ecuadorian H2, KC690929, and KC690922 are sequences from Ae. albopictus collected in tropical climates. We have included sequences from Zhong and collaborators (Zhong et al. 2013), as found in our Supplementary Table S2 and the main text.

We provide another study where these sequences were used as Singapore:

Motoki, M.T., Fonseca, D.M., Miot, E.F. et al. Population genetics of Aedes albopictus (Diptera: Culicidae) in its native range in Lao People’s Democratic Republic. Parasites Vectors 12, 477 (2019). https://doi.org/10.1186/s13071-019-3740-0

Point 3: I find that the analysis of the sequence data made in the manuscript is unhelpful to the conclusions that can be drawn from the data. I really do not see the point using statistics to analyse two haplotypes that are so conserved.

Response 3: We performed several analyses using the sequences data. The genetic diversity of the vector may be associated with the number of introductions, and due to its genetic profile, the mosquito has occupied distinct geographical areas. To compare the genetic diversity of the Ecuadorian populations, a complete set of phylogenetic and population genetic analyses needed to be performed. Using these tests, we have determined the low genetic diversity, the number of populations, association with geographical distance, and other standardized tests.

Point 4: My suggestion is that the paper is rewritten to focus on the fact that several introductions of Aedes albopictus has been made to Ecuador and that the species is spreading in several parts of the country. The climate model presented can be used to indicate that the species has the potential to spread further.

Response 4: We thank the Reviewer's suggestion. However, his suggestion is not practical since the species dispersion is ongoing, highlighting this report's importance. Our results provide the basis for further studies to assess whether the two haplotypes may have differences in insecticide resistance, behavioral changes, and/or vector competition. Failure to carry out this analysis would omit the importance of tracking the insect in Ecuador and neighboring countries that do not yet have the mosquito.

An ecological niche model by Pech-May et al. (2016) estimates that Ae. albopictus will reach places at about 561 masl higher by 2050. Ecuador is a megadiverse country with different climatic zones and ecological niches, including the Galapagos Islands, that are highly vulnerable ecosystems for the arrival of invasive insect populations.